# Impact on Bile Acid Concentrations by Alveolar Echinococcosis and Treatment with Albendazole in Mice

**DOI:** 10.3390/metabo11070442

**Published:** 2021-07-06

**Authors:** Cristina Gómez, Fadi Jebbawi, Michael Weingartner, Junhua Wang, Simon Stücheli, Bruno Stieger, Bruno Gottstein, Guido Beldi, Britta Lundström-Stadelmann, Alex Odermatt

**Affiliations:** 1Division of Molecular and Systems Toxicology, Department of Pharmaceutical Sciences, University of Basel, 4056 Basel, Switzerland; cristina.gomezcastella@unibas.ch (C.G.); fadi.jebbawi@unibas.ch (F.J.); michael.weingartner@unibas.ch (M.W.); simon.stuecheli@unibas.ch (S.S.); 2Department of Infectious Diseases and Pathobiology, Vetsuisse Faculty, Institute of Parasitology, University of Bern, 3012 Berne, Switzerland; junhua.wang@ifik.unibe.ch (J.W.); bruno.gottstein@ifik.unibe.ch (B.G.); britta.lundstroem@vetsuisse.unibe.ch (B.L.-S.); 3Faculty of Medicine, Institute for Infectious Diseases, University of Berne, 3010 Berne, Switzerland; 4Department of Clinical Pharmacology and Toxicology, University Hospital Zurich, University of Zurich, 8091 Zurich, Switzerland; bstieger@ethz.ch; 5Department of Visceral Surgery and Medicine, University Hospital of Berne, 3010 Berne, Switzerland; Guido.Beldi@insel.ch

**Keywords:** bile acid, BSEP, NTCP, alveolar echinococcosis, *Echinococcus multilocularis*, albendazole, LC-MS/MS

## Abstract

Alveolar echinococcosis (AE) caused by *Echinococcus multilocularis* is a chronic, progressive liver disease widely distributed in the Northern Hemisphere. The main treatment options include surgical interventions and chemotherapy with benzimidazole albendazole (ABZ). To improve the current diagnosis and therapy of AE, further investigations into parasite–host interactions are needed. This study used liquid chromatography–tandem mass spectrometry (LC-MS/MS) to assess serum and liver tissue bile acid profiles in the *i.p.* chronic *E. multilocularis*-infected mouse model and evaluated the effects of the anthelmintic drug ABZ. Additionally, hepatic mRNA and protein expression of enzymes and transporters regulating bile acid concentrations were analyzed. AE significantly decreased unconjugated bile acids in serum and liver tissue. Taurine-conjugated bile salts were unchanged or increased in the serum and unchanged or decreased in the liver. Ratios of unconjugated to taurine-conjugated metabolites are proposed as useful serum markers of AE. The expression of the bile acid synthesis enzymes cytochrome P450 (CYP) 7A1 and aldo-keto reductase (AKR) 1D1 tended to decrease or were decreased in mice with AE, along with decreased expression of the bile acid transporters Na+/taurocholate cotransporting polypeptide (NTCP) and bile salt efflux pump (BSEP). Importantly, treatment with ABZ partially or completely reversed the effects induced by *E. multilocularis* infection. ABZ itself had no effect on the bile acid profiles and the expression of relevant enzymes and transporters. Further research is needed to uncover the exact mechanism of the AE-induced changes in bile acid homeostasis and to test whether serum bile acids and ratios thereof can serve as biomarkers of AE and for monitoring therapeutic efficacy.

## 1. Introduction

Human alveolar echinococcosis (AE) caused by *Echinococcus multilocularis* is a severe liver disease with high morbidity and poor prognosis when handled inappropriately [1,2]. In the Northern Hemisphere, 0.3 to 3 per 1,000,000 inhabitants get infected with *E. multilocularis* annually, with the numbers increasing [1,2,3]. Adult stages of the tapeworm are mainly noted in the intestine of red and arctic foxes, although domestic dogs and cats can also act as definitive hosts [3,4]. A prevalence of <1.5% was found in privately owned rural and urban pet dogs, whereas it was between 3% and 8% in dogs with free access to rodent habitats, such as farm dogs and hunting dogs. The course of the infection is characterized by a long-term, primarily intrahepatic growth of the metacestode (larval stage) in the intermediate host, including small mammals such as mice [5]. In humans, invasion of the bile ducts by *E. multilocularis* leads to cholangitis and portal hypertension, and the disease can progress to liver cirrhosis after a long latent, asymptomatic period [5,6].

A characteristic feature of AE is the tumor-like growth of the metacestode, which may lead to infiltration of neighboring organs. The only curative treatment is surgical resection of the metacestode tissue, supported by pre- and post-operative chemotherapy [2]. Treatment in inoperable AE cases consists of a long-term and often life-long administration of benzimidazoles albendazole (ABZ) or mebendazole. Long-term treatment with benzimidazoles is required because these drugs are parasitostatic but not parasiticidal, and adverse reactions, including severe hepatotoxicity, are frequently observed [2,7,8].

Although it is well known that impaired liver function in situations such as cholestasis and cirrhosis or in drug-induced liver injury can have profound effects on bile salt homeostasis, little is known whether serum or intrahepatic bile salts are altered during *E. multilocularis* infection and whether this is influenced by the ABZ treatment.

Primary bile acids are synthesized from cholesterol by a complex multistep pathway in hepatocytes [9,10]. The so-called classic (or neutral) pathway starts by introducing a hydroxy-group at the 7α-position of cholesterol, catalyzed by CYP7A1, which constitutes the rate-limiting step of hepatic bile acid biosynthesis. CYP7A1 expression and activity is controlled by various transcription factors and feedback mechanisms. The alternative (or acidic) pathway is initiated by the side chain hydroxylation of cholesterol through sterol 27-hydroxylase (CYP27A1) and is primarily extrahepatic. The primary di- and trihydroxy bile acids are conjugated with taurine or glycine and exported from hepatocytes across the canalicular membrane by the bile salt export pump (BSEP) into the canaliculi, from where they drain with bile into the upper small intestine. The ratio of dihydroxy to trihydroxy bile acids as well as the ratio of glycine to taurine conjugates are different between species and hence result in species-specific bile salt pool compositions [11]. Bile salts are the main organic component of bile [12,13].

Most of the bile acids are absorbed by active transport in the distal ileum and transported back to the liver via the portal vein, while some part is metabolized by the gut flora into secondary bile acids [9,10,14]. In the liver, bile salts are transported from the portal blood plasma across the basolateral membrane on hepatocytes by numerous transport proteins. While Na^+^-taurocholate cotransporting polypeptide (NTCP) predominantly transports bile salts and organic anion transporters (OATPs) predominantly mediate the uptake of bile acids from the blood plasma through the basolateral membrane into the hepatocytes [15,16,17,18,19], BSEP represents the rate-limiting step in the secretion of bile salts into the bile fluid. Inherited and acquired forms of liver diseases, which impair proper BSEP function, can lead to intracellular accumulation of bile acids in hepatocytes, i.e., cholestasis. If persistent, cholestasis can cause hepatocellular damage or even cell death [16,17,19]. Elevated hepatocellular levels of bile acids lead to the activation of the farnesoid X receptor (FXR), which directly downregulates CYP7A1 expression and indirectly the sinusoidal uptake transporter NTCP, but upregulates the efflux transporter BSEP to reduce the concentration of intracellular bile salts [16,17,19]. FXR activation upon hepatic bile acid accumulation also increases sinusoidal bile acid efflux by inducing the expression of multidrug resistance-associated protein (MRP)4 and heterodimeric organic solute transporters (OST)α/OSTβ (SLC51A/SLC51B) [14,15,18,20,21].

In many hepatic and intestinal diseases, serum bile acid concentrations are altered due to impaired hepatic synthesis and metabolism and/or intestinal absorption. Thus, serum bile acid concentrations may serve as prognostic and diagnostic markers of liver dysfunctions and diseases [22,23,24,25,26]. Consequently, the present study assessed changes in serum and liver tissue bile acid profiles in mice infected *i.p.* with metacestodes of *E. multilocularis* using ultra-high performance liquid chromatography–tandem mass spectrometry (LC-MS/MS). Furthermore, the effect of the currently most frequently used treatment, i.e., ABZ, on the serum and intrahepatic bile acid profiles was assessed. Additionally, the mRNA and protein expression levels of key bile salt transporters and enzymes involved in bile acid synthesis were studied in liver tissues.

## 2. Results

### 2.1. Murine Model of Alveolar Echinococcosis

In the present investigation, we analyzed serum and liver tissue samples from a previous study on innate and adaptive immune responses in mice infected *i.p.* with metacestodes of *E. multilocularis* and treated with vehicle or ABZ [27]. The natural mode of infection (or primary infection) of intermediate hosts with *E. multilocularis* represents peroral uptake of infectious eggs, which also includes the first phase of intrahepatic parasite development. However, such a model is rarely applied in experimental infection due to the limited availability of infectious eggs for experimental infection, the high risk for the experimenter, and the necessity of high biosafety precautions. The most commonly applied, secondary murine AE infection model includes *i.p.* inoculation of *E. multilocularis* metacestode tissue suspension, mimicking a chronic infection. In this model, the parasite mostly grows in the peritoneal cavity. A macroscopic and histological assessment in an earlier study [27] thus did not detect any visible parasitic structures within the livers of any mouse, but inflammatory cell infiltrates with elevated levels of pro-inflammatory cytokines were observed in infected mice. Furthermore, the treatment efficacy of ABZ was demonstrated in the previous study by a significantly reduced parasite weight (isolated from the peritoneal cavity), the histopathology (immune cell infiltration), and reduced liver tissue cytokine levels.

### 2.2. Bile Acid Profiles

To assess the potential effects of *E. multilocularis* infection and its main treatment ABZ on the serum bile acid profiles in mice, a recently established LC-MS/MS-based method was applied to quantify a series of conjugated and unconjugated bile acids [28]. Glycine-conjugated bile acids were at or below the lower limit of detection of the quantification method and are not listed in Table 1. The mean concentrations of each individual bile acid as well as the sums of taurine-conjugated, unconjugated, and total bile acids analyzed for the four treatment groups are listed in Table 1, and data for individual bile acids are shown in Appendix A. The concentrations of all unconjugated primary (CA, CDCA, αMCA, βMCA, and ωMCA) and secondary (UDCA, DCA, HDCA, 7oxoDCA, and 12oxoLCA) bile acids were 4.6-fold lower in the AE group compared with the control group. In contrast, the taurine-conjugated bile salts (TCDCA, TUDCA, TDCA, TαMCA, and TωMCA) were either not altered or significantly increased (TCA and TβMCA) in AE compared to non-infected control (CTRL) mice, with 1.5-fold higher levels of total taurine-conjugated bile acids in the AE group. The total circulating bile acids were 2.5-fold lower in the AE compared to CTRL mice.

In addition, bile acids were quantified in the liver tissue using the previously described LC-MS/MS method [28]. The mean concentrations of each individual bile acid in addition to the sums of unconjugated, conjugated, and total bile acids analyzed for the four treatment groups are listed in Table 2, and the data for individual bile acids are depicted in Appendix A. The concentrations of all unconjugated bile acids were decreased from 2- to 60-fold in the AE group compared with the control group. CDCA, UDCA, HDCA, and 7oxoDCA had lower values than the limit of detection of the method. The taurine-conjugated bile salts were either not altered or significantly decreased (TUDCA and TαMCA) in AE compared to AE-ABZ mice.

ABZ is the drug of choice to treat AE in humans [1,2]. In this mouse model of AE, ABZ treatment was well tolerated. No significant changes in any of the serum or liver tissue bile acids analyzed were detected in the CTRL-ABZ compared to the CTRL group (Table 1 and Table 2, Appendix A), although there was a trend to decreased total unconjugated bile acids and taurine-conjugated bile salts in liver tissues of CTRL-ABZ mice. Importantly, ABZ treatment of *E. multilocularis*-infected mice partially or completely reversed the changes in serum and liver tissue bile acid concentrations observed in AE mice (compared to AE-ABZ), and the bile acid concentrations of all analyzed metabolites in the AE-ABZ group were comparable to those of the non-infected CTRL group. A comparison of the relative abundance of each individual bile acid in percentage shows highly similar profiles for CTRL, CTRL-ABZ, and AE-ABZ but a clearly distinct profile for the AE group (Figure 1). The graphical representation of the profiling facilitates distinguishing the different treatment groups (Figure 1A for serum profiles and Figure 1B for liver tissue profiles).

As ratios often show lower inter-individual variations than single analytes, they may represent more robust markers to detect changes in bile acid homeostasis. The ratio of total taurine-conjugated to unconjugated bile acids as well as the ratios of TCA/CA, TαMCA/αMCA, and TβMCA/βMCA were all approximately 10-fold higher compared to the CTRL group in AE mice, followed by complete reversal upon ABZ treatment in the serum samples (Figure 2A and Appendix A, Table 3). The intrahepatic ratios tended to be higher in the AE group compared to the CTRL and the AE-ABZ groups (Figure 2B and Appendix A, Table 3).

### 2.3. Effect of AE and ABZ Treatment on Enzymes Involved in Bile Acid Synthesis

To assess whether AE and/or the treatment with ABZ affect bile acid synthesis, the mRNA levels of the rate-limiting enzyme Cyp7a1, the sterol 27-hydroxylase Cyp27a1, which initiates the acidic pathway or alternative pathway for bile acid synthesis, and of the 5β-reductase Akr1d1 were analyzed by quantitative PCR (qPCR). The expression of Cyp7a1, and to a lesser extent that of Cyp27a1, tended to be decreased in the liver tissues of *E. multilocularis*-infected mice (Figure 3). These trends were reversed by treatment with ABZ. Attempts to determine the protein expression of these enzymes in mouse liver tissues failed as no suitable antibodies and conditions could be identified. Regarding the 5β-reductase Akr1d1, significantly lower mRNA levels along with reduced protein levels were observed in liver tissues from AE mice compared to CTRL, and ABZ treatment reversed the decreased mRNA and protein expression.

### 2.4. Influence of AE and ABZ Treatment on the Expression of Bile Acid Transporters

Besides enzymes involved in bile acid synthesis and metabolism, several transport proteins regulate the composition and concentrations of bile acids in the blood, liver tissue, and bile fluid. To investigate a possible involvement of bile acid transporters in the observed alteration of the bile acid profile upon *E. multilocularis* infection, the mRNA levels of Bsep, Ntcp, Ostα, Ostβ, Mrp2, Mrp4, Oatp1a1, and Oatp1b2 were quantified by qPCR. A trend of decrease in Bsep mRNA levels and significantly lower Ntcp mRNA levels were observed in the liver tissues of *E. multilocularis*-infected mice (Figure 4, lower right panel in A and B). Furthermore, a decrease in Ostα and Oatp1b2 mRNA levels and a trend of decrease in Mrp2 mRNA expression was found in the AE group (Appendix A). These effects were fully reversed by ABZ treatment, and ABZ itself had no direct effect on any of the genes of bile acid transporters measured. The mRNA levels of Oatp1a1, Mrp4, and Ostβ were not different between the treatment groups.

For the assessment of protein expression using western blot analysis, antibodies suitable for BSEP, NTCP, and OATP1A1 were available. Protein expression was analyzed by densitometry and normalized to the house-keeping protein LMNB1. The protein expression levels of the bile acid transporters BSEP and NTCP were both significantly lower in the *E. multilocularis*-infected mice (AE group) compared to the CTRL mice (Figure 4A,B). In contrast, OATP1A1 expression was not significantly altered in any of the four treatment groups (Figure 4C).

## 3. Discussion

In humans, AE affects the liver as the primary site and progresses by continuous infiltrative proliferation, thereby destroying liver tissue, invading the adjacent organs, and metastasizing to the lungs and the brain [1,6]. The disease progression of AE is usually slow accompanied with underlying inflammation and often not diagnosed until it is well advanced [6,29]. The evaluation of serum bile acid concentrations, including the determination of bile acid profiles as well as ratios of certain bile acids (e.g., taurine-conjugated/unconjugated, CA/CDCA) have the potential as diagnostic markers to distinguish different liver diseases and monitor disease progression and therapeutic efficacy [22,23,24,26].

For the present study, we used an animal model for chronic infection with an *i.p.* application of metacestodes of *E. multilocularis* to mice. While this model does not show macroscopic changes in livers of infected animals, a diffuse infiltration of immune cells in livers and elevated pro-inflammatory cytokines in the plasma of treated animals is found [27]. Hence, while the animal model does not reproduce completely, the human pathology of AE infection and an oral infection model would more accurately represent it. Our *i.p.* model clearly reproduces the inflammatory component of this disease while avoiding the higher biosafety considerations and limited availability of infections eggs of an oral infection model. The decreased unconjugated bile acids together with unaltered or slightly elevated taurine-conjugated bile salts in plasma and decreased intrahepatic unconjugated bile acid levels and mildly reduced intrahepatic taurine-conjugated bile salts exclude cholestasis in our diseased animals but support a decreased bile salt pool. We can, therefore, not completely rule out that the systemic inflammatory mediators (also seen in other forms of liver disease) contribute on top of inflammatory processes in liver to the alterations in bile acid and bile salt homeostasis in infected mice. This question needs further investigation, including the determination of bile salt pools in the different groups. The ratio of total unconjugated to total taurine-conjugated metabolites as well as that of TCA/CA and TβMCA/βMCA in serum are proposed as useful markers to detect the effect of AE and monitor the efficacy of ABZ treatment in the murine chronic infection model applied. It needs to be noted that in this model of secondary infection, histological analysis did not show any macroscopic changes in the liver, but infiltration of immune cells and elevated pro-inflammatory cytokines were observed [27].

Common symptoms observed in AE patients at a late stage of disease, where hepatic lesions are seen in histological analysis, include jaundice, abdominal pain, and weight loss [6,29,30]. Studies in patients with jaundice showed an altered bile acid metabolism, with an increase in the circulating levels of conjugated bile salts and an elevated ratio of taurine-conjugated to unconjugated bile acids in symptomatic patients [26,31]. The utility of the serum bile acid ratio markers mentioned above should be further studied in animal models, i.e., in mice perorally infected with *E. multilocularis* eggs and at different time points following infection, as well as in humans.

Gene expression analyses of the present study indicated a decreased hepatic bile acid synthesis by lower CYP7A1 and AKR1D1 expression levels, along with decreased canalicular biliary secretion via BSEP, reduced bile acid and bile salt uptake from the portal circulation by NTCP and OATP1B2, and reduced efflux to the general circulation by OSTα and MRP4 (Figure 5). An inhibition of the above-mentioned bile acid transporters may be a result of the hepatic inflammation, shown in the previous study [27], leading to reduced FXR-mediated activation of SHP and PPARα that are involved in the regulation of the expression of these bile acid transporters [14]. A rapid reduction of bile formation via downregulation of both basolateral bile acids uptake (NTCP) and canalicular efflux system (BSEP) have been observed in response to inflammation [16,20]. A decreased activity of BSEP and OSTα-OSTβ directly relates to decreased bile acid-dependent activation of FXR signaling, which can lead to liver injury [14,18]. The role of inflammation on the bile acid pool in the AE model applied needs further investigations, including analysis of bile flow and bile acid excretion by the kidney and feces.

In other conditions, such as chronic forms of cholestasis, downregulation of NTCP and upregulation of basolateral bile acid export systems were observed (MRP4 and OSTα-OSTβ) [15,16]. This contrasts the present study on AE, where OSTα and MRP4 were decreased. Moreover, if FXR activity is compromised, one would expect an elevated expression of the rate-limiting bile acid synthesis enzyme CYP7A1. In the liver, TCA, acting as FXR agonist, tended to be lower, while TβMCA, an FXR antagonist [32], was not altered, not supporting profound effects through altered presence of FXR ligands. Unfortunately, bile fluid was not collected, which may provide further mechanistic insight in a follow-on study. Further studies are needed to uncover the mechanism underlying the altered bile acid homeostasis in AE and to elucidate its consequences for AE progression.

Evidence from earlier studies of helminth infections with the liver as the primary affected site indicates an important role of bile acids in the modulation of helminth–host interactions. The liver fluke *Opisthorchis viverrini*, for example, resides in the biliary tree in an environment of very high bile acid concentrations, and a study in infected hamsters found elevated levels of the secondary bile acid DCA [33], suggesting disturbed activity of the microbiome and/or altered intestinal reuptake of bile acids. Another example includes *Echinococcus granulosus*, where the development of the larvae into secondary hydatid cysts is promoted by bile acids and high levels of bile acids are needed for the development of adult worms [34]. Interestingly, a serum metabolome analysis in Beagle dogs infected with *Toxocara canis* showed that the bile acid CA was increased 24 h post-infection but decreased 10 days after infection, along with a pronounced increase in TCA and TCDCA, thus indicating a shift from unconjugated bile acids to taurine-conjugated bile salts [35], similar to the observed effect in the present study in AE. Furthermore, a study in mice infected with cysticerci of *Taenia crassiceps* showed altered hepatic metabolism with enhanced production of taurine and glycine [36]. Although that study did not assess serum bile acids and salts, the enhanced production of these amino acids suggests an increased capacity for bile acid conjugation. These observations may provide an explanation for the observed shift from conjugated to taurine-conjugated bile acids in AE in the present study, with a potentially higher rate of taurine-conjugation in infected livers. Besides an altered hepatic bile acid metabolism and transport, parasite infections can also disturb intestinal activity and bile acid transport. For example, mice infected with *Trichinella spiralis* displayed gut dysfunction with a decreased bile acid reuptake in the ileum, suggesting an enhanced fecal loss of bile acids [37]. Thus, follow-on experiments should investigate the role of inflammation, microbial bile acid metabolism, bile flow, and intestinal bile acid transport in AE models. Furthermore, it will be important to study the role of FXR, as agonists of this receptor were found to exert protective effects on microbial infections [38,39,40].

Additionally, the present work showed that ABZ treatment ameliorates *E. multilocularis* infection and almost completely reverses the effects on serum bile acids and gene expression. This further demonstrates the efficacy of ABZ, mainly due to inhibition of parasite proliferation and immune cell infiltration as reported previously [27]. ABZ treatment itself tended to decrease hepatic bile acid concentrations. Whether this may contribute to the hepatotoxic effects seen upon long-term treatment with this drug [41] remains to be investigated. Ultimately, an improved understanding of the pathways involved in the disturbed bile acid homeostasis may help designing novel therapeutic strategies to combat AE in humans, either alone or in combination with benzimidazoles such as ABZ.

## 4. Materials and Methods

### 4.1. Chemicals and Reagents

Ultrapure water was obtained using a Milli-Q^®^ Integral 3 purification system equipped with an EDS-Pak^®^ Endfilter for the removal of endocrine active substances (Merck Millipore, Burlington, MA, USA). Acetonitrile (HPLC-S Grade) was purchased from Biosolve (Dieuze, France), methanol (CHROMASOLV™ LC-MS grade) from Honeywell (Charlotte, NC, USA), isopropanol (EMSURE^®^ for analysis) from Merck Millipore, and formic acid (Puriss. p.a. ≥98%) from Sigma-Aldrich (St. Louis, MO, USA). Bile acids and internal standards were purchased from Sigma-Aldrich or Steraloids (Newport, RI, USA) as described recently [28].

### 4.2. Mice

Female C57BL/6 mice (8 weeks old, n = 24) were housed under standard conditions in a conventional daylight/night cycle room. The mice were fed a standard pellet chow and water ad libitum. All experiments were performed in accordance with the Federation of European Laboratory Animal Science Association (FELASA) guidelines. The experiments performed in this animal study were reviewed and accepted by the cantonal veterinary authority of the Canton of Bern, Switzerland, and were performed in agreement with the guidelines for care and use of laboratory animals (license BE-112/17).

The mice were randomly divided into four groups: (1) non-infected control (CTRL, n = 6), (2) *E. multilocularis* infected (AE, n = 6), *E. multilocularis* infected treated with albendazole (AE-ABZ, n = 6), and (4) non-infected treated with ABZ (CTRL-ABZ, n = 6). The animals were examined daily for their health status and changes in weight during the experimental period. All animal experiments were conducted within a laminar flow safety hood. At the end of the experimental part, the mice were euthanized using CO_2_. The blood was collected and the serum was separated by centrifugation at 3000× *g* for 15 min. The serum samples were kept at −80 °C for later analysis. Liver tissue was immediately collected, frozen in liquid nitrogen, and stored at −80 °C for later analysis.

Infection with *E. multilocularis* metacestodes was performed by *i.p.* injection [27]. Briefly, *E. multilocularis* (isolate H95) was extracted and maintained by serial passages in C57BL/6 mice. Aseptic removal of infectious material from the abdominal cavity of previously infected animals was used for propagation of AE in mice. The collected tissue was grinded through a sterile 50 μm filter, roughly 100 vesicular cysts were suspended in 100 μL sterile PBS and administrated via *i.p.* injection to groups 2 (AE) and 3 (AE-ABZ). The mice of control groups 1 (CTRL) and 4 (CTRL-ABZ) received 100 μL of sterile PBS. The ABZ treatment started after 6 weeks of initial infection. The mice of groups 1 (CTRL) and 2 (AE) were administered 100 μL corn oil and those of groups 3 (AE-ABZ) and 4 (CTRL-ABZ) received 100 μL ABZ in corn oil (200 mg/kg mouse/injection) orally five times per week. The treatment was terminated after 8 weeks by euthanizing the examined animals. The serum and liver tissue samples were used from a previous study [27].

### 4.3. Quantification of Bile Acids in Serum

Bile acids were quantified as described recently [28]. Briefly, 10 µL of serum were diluted 1:4 with water, followed by adding 900 µL of 2-propanol for protein precipitation and a mixture of deuterated internal standards. Extraction was performed by continuous shaking for 30 min at 4 °C and centrifugation at 16,000× *g* for 10 min. Supernatants were transferred to new tubes, evaporated to dryness, and reconstituted with 100 µL of methanol to water (1:1, *v*/*v*). Samples were analyzed by LC-MS/MS, consisting of an Agilent 1290 UPLC coupled to an Agilent 6490 triple quadrupole mass spectrometer equipped with an electrospray ionization (ESI) source (Agilent Technologies, Basel, Switzerland). Chromatographic separation of the bile acids was achieved using reversed-phase column (ACQUITY UPLC BEH C18, 1.7 mm, 2.1 µm, 150 mm, Waters, Wexford, Ireland).

### 4.4. Quantification of Bile Acids in Liver Tissue

Bile acid extraction from liver tissue samples has been described previously [28]. Briefly, liver tissue (20 ± 5 mg) was homogenized in 900 µL of chloroform:methanol:water (1:3:1, *v*/*v*/*v*) and 100 µL internal standard mixture using a Precellys tissue homogenizer (Bertin Instruments, Montigny-le-Bretonneux, France) with three cycles of 30 s at 6500 rpm with 30 s break between cycles. The samples were centrifuged (10 min, 20 °C, 16,000× *g*) and supernatant was transferred to a new tube. Another 800 µL of extraction solvent were added to the samples and the process was repeated. Combined supernatant (1600 µL) was evaporated using a Genevac EZ-2 (SP Scientific, Warminster, PA, USA) and the dried samples resuspended in 200 µL methanol:water (1:1, *v*/*v*). Then, 3 µL was injected into a liquid chromatrography–tandem mass spectrometry system, consisting of an Agilent 1290 UPLC coupled to an Agilent 6490 triple quadrupole mass spectrometer with an electrospray ionization source (Agilent Technologies, Basel, Switzerland). The bile acid analytes were separated using a reversed-phase column (Acquity UPLC BEH C18, 1.7 mm, 2.1 mm, 150 mm; Waters, Wexford, Ireland).

### 4.5. Total RNA Extraction and qPCR

Total RNA was isolated from the liver tissues using the Qiagen RNeasy MiniKit and QIAcube instrument according to the manufacturer’s protocol (SABioscience, Frederick, MD, USA). The quality and concentration of RNA was determined using a Nanodrop™ one C (Cat#13-400-519, Thermo Fisher Scientific, Waltham, MA, USA). Only samples with a 260 nm to 280 nm ratio between 1.9 and 2.1 and a 260 nm to 230 nm ratio between 1.5 and 2.0 were further processed. cDNA was synthesized using GoScript Reverse Transcriptase (Cat#A5003, Promega, Madison, WI, USA). The KAPA SYBR Fast Kit (Cat# SFUKB, Merck, Darmstadt, Germany) was used for qPCR analysis, and the reactions were performed on a Rotor Gene real-time cycler (Corbett Research, Sydney, New South Wales, Australia). The data were normalized to the expression levels of the endogenous control gene β-actin. The primers are listed in Appendix A.

### 4.6. Protein Expression/Western Blot

Approximately 7 mg of frozen liver tissues were homogenized (6500 rpm, 30 s, 4 °C, Precellys 24 tissue homogenizer) in 450 µL RIPA buffer (50 mM Tris-HCl, pH 8.0, with 150 mM sodium chloride, 1.0% NP-40, 0.5% sodium deoxycholate, and 0.1% sodium dodecyl sulfate) containing protease inhibitor cocktail (Cat#11836153001, Merck, Darmstadt, Germany) and centrifuged (4 min, 4 °C, 16,000× *g*). Protein concentration in supernatants was measured using standard bicinchoninic acid assay (Pierce BCA Protein Assay Kit, Thermo Fisher Scientific). The samples were heated (5 min at 95 °C) in Laemmli solubilization buffer (LSB; 60 mM Tris-HCl, 10% glycerol, 0.01% bromophenol blue, 2% sodium dodecyl sulfate, pH 6.8, and 5% β-mercaptoethanol) and 20 μg of total protein were separated by 8–14% SDS-PAGE and transferred to PVDF membranes (Immobilon-P Membran, PVDF, pore size: 0.45 µm). The membranes were blocked (1 h, 25 °C) in TBST containing 5% nonfat dry milk (5% nonfat dry milk powder in 20 mM Tris-HCl with 0.1% Tween-20) or 1% bovine serum albumin (in 20 mM Tris-HCl with 0.1% Tween-20). AKR1D1 protein expression was determined using mouse monoclonal anti-AKR1D1 antibody (1:1000, 4 °C, overnight). The washed membranes were incubated with HRP-conjugated secondary goat anti-mouse antibody (1:4000, 25 °C, 1 h). BSEP protein expression was analyzed using a rabbit polyclonal anti-BSEP antibody [42] (1:4000, 4 °C, overnight). The membrane was washed and incubated with HRP-conjugated secondary goat anti-rabbit antibody (1:4000, 25 °C, 1 h). OATP1A1 protein expression was determined using a rabbit polyclonal anti-OATP1A1 antibody [43] (1:1000, 4 °C, overnight) and HRP-conjugated secondary goat anti-rabbit antibody (1:4000, 25 °C, 1 h). Protein levels of NTCP were measured using rabbit polyclonal anti-NTCP antibody [44] (1:1000, 4 °C overnight) and HRP-conjugated secondary goat anti-rabbit antibody (1:4000). LMNB1 served as loading control and was detected using rabbit monoclonal anti-LMNB1 antibody (1:1000, 4 °C, overnight) followed by HRP-conjugated secondary goat anti-rabbit antibody (1:2000, 25 °C, 1 h). Protein bands were visualized by Immobilon Western Chemiluminescence HRP substrate and semi-quantitatively analyzed by densitometry, normalized to LMNB1 protein levels, using Image J software (RRID:SCR_003070, version 1.53n).

### 4.7. Data Analysis and Statistics

For LC-MS/MS data, MassHunter Acquisition Software (Agilent Technologies, Inc., Santa Clara, CA, USA) and MassHunter Quantitative Analysis vB.07.01 (Agilent Technologies, Inc., Santa Clara, CA, USA) were used for quantification. The Kruskal–Wallis test and Dunn’s multiple comparison were used to analyze significance of differences between groups. Grubbs’ test was performed to determine outliers. Statistical significance was established at *p* < 0.05. Statistical analysis and graphs were performed using GraphPad Prism v5.02 (GraphPad Software, Inc., San Diego, CA, USA).

## 5. Conclusions

This work applied LC-MS/MS to quantify bile acids in serum and liver tissue samples of mice infected with *E. multilocularis* in the absence or presence of the parasitostatic drug ABZ. The results revealed decreased unconjugated bile acids and unchanged or increased taurine-conjugated bile salts in the serum, suggesting the use of ratios of unconjugated to taurine-conjugated metabolites as markers for AE and to monitor therapeutic efficacy. Gene expression analyses showed decreases in bile acid synthesis enzymes and in key transport proteins. ABZ, which did not substantially affect bile acid homeostasis itself, reversed the observed effects on serum and liver tissue bile acids and on gene expression. Follow-on studies need to uncover the exact mechanism underlying the observed effects and to evaluate the bile acid ratio markers in AE and its treatment.

## Figures and Tables

**Figure 1 metabolites-11-00442-f001:**
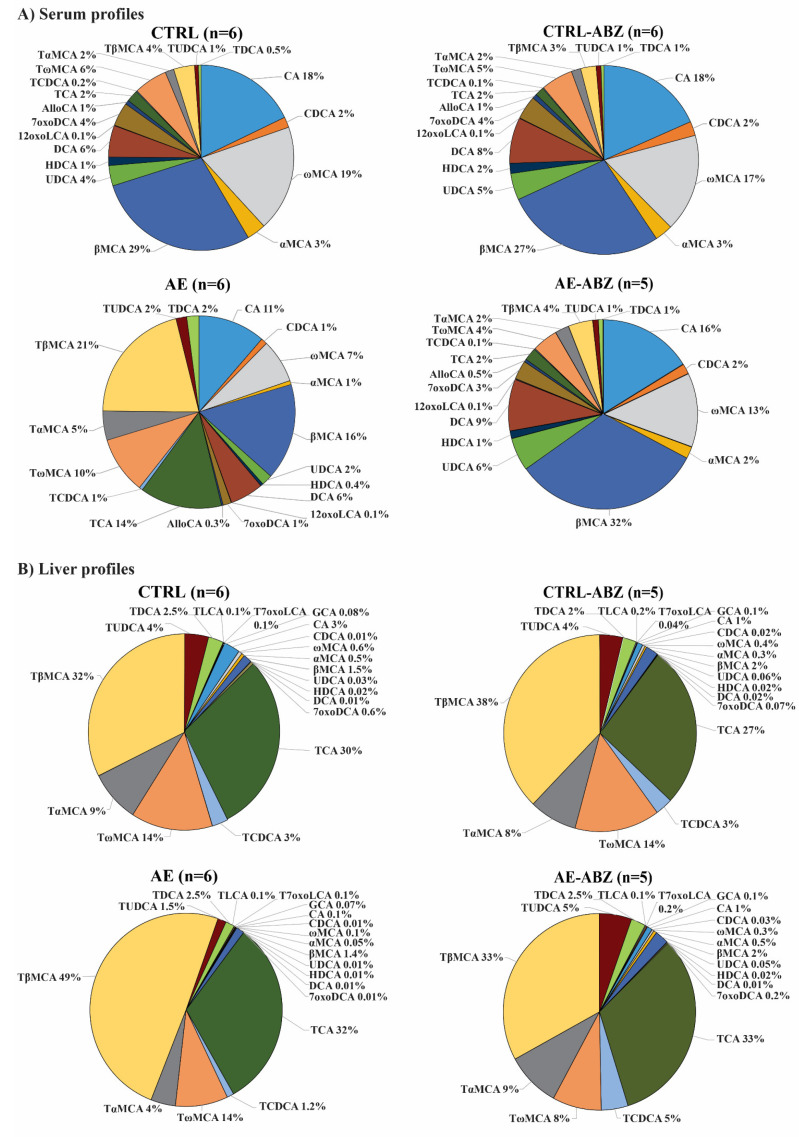
Bile acid profiles of the four different mouse groups in serum (**A**) and in liver tissue (**B**). The relative amounts of individual bile acids, indicated by different colors, are shown for the four different groups: non-infected control (CTRL, n = 6), *E. multilocularis*-infected (AE, n = 6), non-infected and ABZ-treated control (CTRL-ABZ, n = 6 for serum, n = 5 for liver tissue due to exclusion of an outlier), and *E. multilocularis*-infected and ABZ-treated mice (AE-ABZ, n = 5, one outlier with aberrant concentrations was removed). Data represent relative abundances (%) of individual bile acids.

**Figure 2 metabolites-11-00442-f002:**
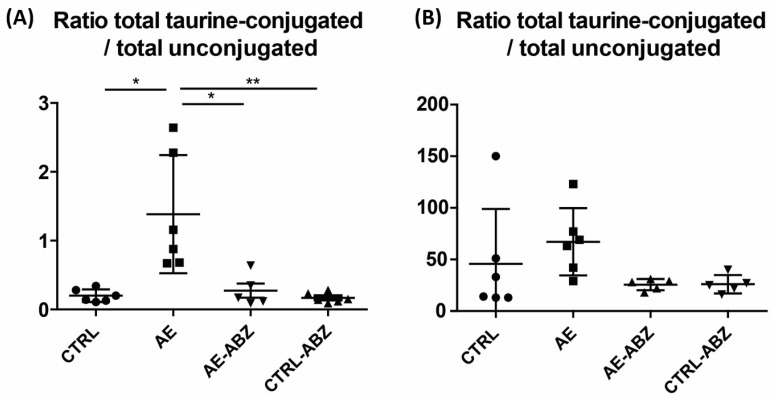
Increased ratio of taurine-conjugated to unconjugated bile acids in AE. The sum of the taurine-conjugated bile acids divided by the sum of unconjugated bile acids was determined in the four different treatment groups, in serum samples (**A**) and liver tissues (**B**). Non-infected control (CTRL, n = 6), *E. multilocularis*-infected (AE, n = 6), *E. multilocularis*-infected and ABZ-treated (AE-ABZ, n = 5, one outlier with aberrant concentrations was removed), and non-infected and ABZ-treated control mice (CTRL-ABZ, n = 6 in serum and n = 5 in liver tissue due to exclusion of an outlier). Values are expressed as mean ± SD. * *p* < 0.05 and ** *p* < 0.01.

**Figure 3 metabolites-11-00442-f003:**
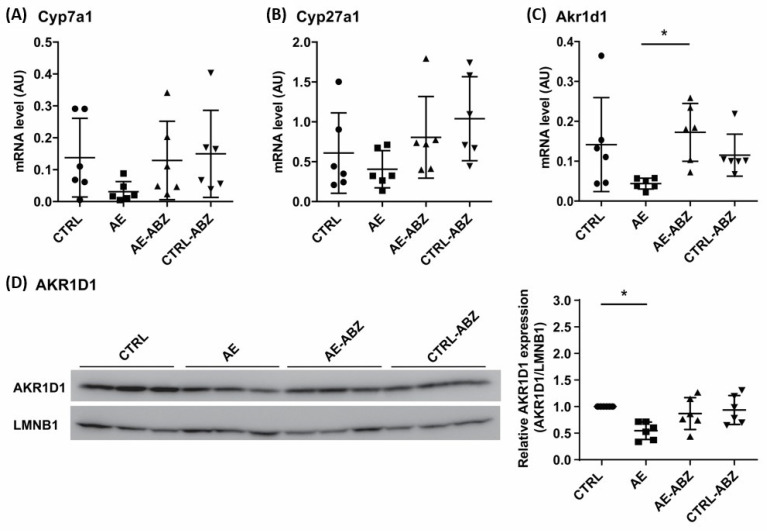
Expression levels of enzymes involved in bile acid synthesis. The mRNA levels of Cyp7a1 (**A**), Cyp27a1 (**B**), and Akr1d1 (**C**) and the protein levels of AKR1D1 (**D**) were determined in the liver tissues of non-infected control (CTRL, n = 6), *E. multilocularis*-infected (AE, n = 6), *E. multilocularis*-infected and ABZ-treated (AE-ABZ, n = 6), and non-infected and ABZ-treated control mice (CTRL-ABZ, n = 6). One representative blot (of two) containing samples from three different mice is shown in (**D**) on the left and densitometry results are shown on the right, representing data from the two blots on samples from six mice, normalized to lamin B1 (LMNB1) control and with CTRL set as 1. Values are expressed as mean ± SD. * *p* < 0.05.

**Figure 4 metabolites-11-00442-f004:**
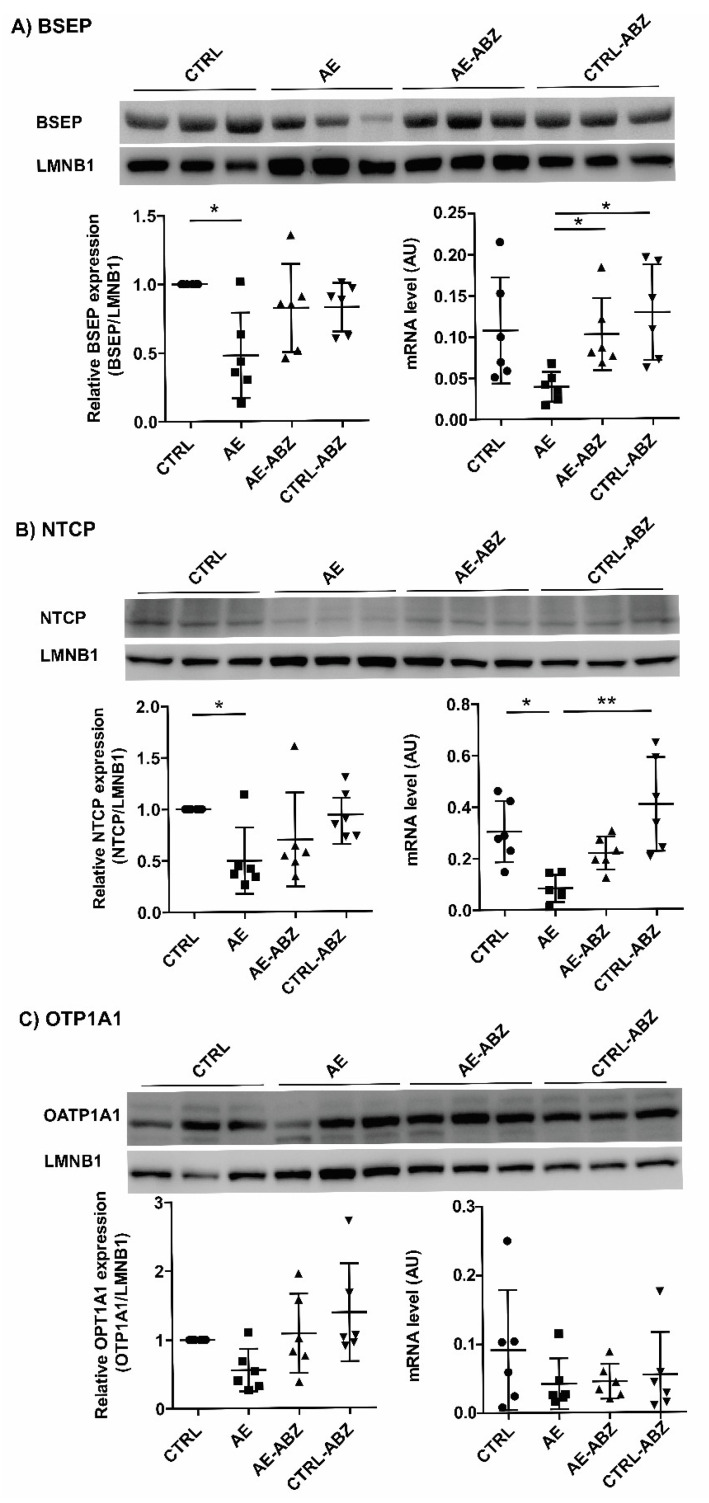
Hepatic mRNA and protein expression levels of BSEP, NTCP, and OATP1A1 in the four treatment groups. Western blot and semi-quantitative analysis by densitometry of protein levels of transporters BSEP (**A**), NTCP (**B**), and OATP1A1 (**C**) in the liver tissues of non-infected control (CTRL, n = 6), *E. multilocularis*-infected (AE, n = 6), *E. multilocularis*-infected and ABZ-treated (AE-ABZ, n = 6), and non-infected and ABZ-treated control mice (CTRL-ABZ, n = 6). One representative blot (of two) containing samples from three different mice is shown in the top panel. Densitometry results represent data from the two blots on samples from six mice (mean ± SD), normalized to lamin B1 (LMNB1) control, and with CTRL set as 1. * *p* < 0.05 and ** *p* < 0.01.

**Figure 5 metabolites-11-00442-f005:**
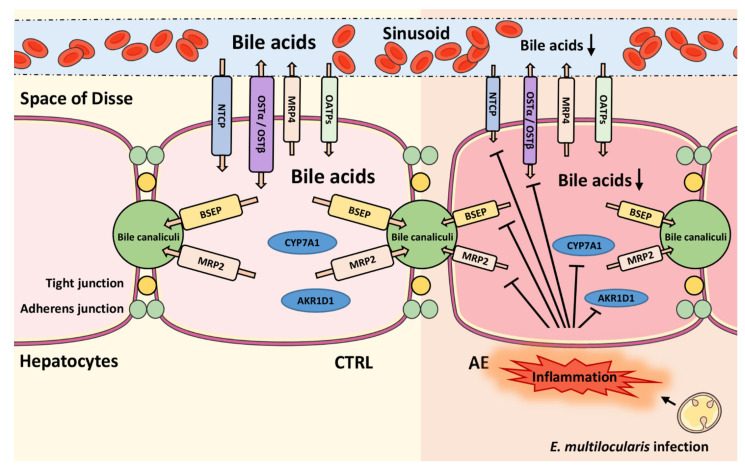
Schematic overview of key transport proteins involved in hepatic bile acid homeostasis. Left panel: situation in hepatocytes of control mice; right panel: model of the situation in hepatocytes in experimental murine alveolar echinococcosis (AE). The *i.p.* infection results in hepatic inflammation, suppressing the expression of the bile acid biosynthesis enzyme cytochrome P450 (CYP) 7A1 and aldo-keto reductase (AKR) 1D1 and of the bile salt transporters bile salt efflux mump (BSEP), multidrug resistance-associated protein (MRP) 2, Na+/taurocholate cotransporting polypeptide (NTCP), and organic solute transporters (OST)α (SLC51A) and OSTβ (SLC51B). OATPs, organic anion transporters; CTRL, control.

**Table 1 metabolites-11-00442-t001:** Effects of *E. multilocularis* infection and ABZ treatment on bile acids in the serum of mice. The serum was collected from non-infected control (CTRL, n = 6), *E. multilocularis*-infected (AE, n = 6), *E. multilocularis*-infected and ABZ-treated (AE-ABZ, n = 5, one outlier with aberrant concentrations was removed), and non-infected and ABZ-treated control mice (CTRL-ABZ, n = 6). * *p* < 0.05 AE vs. CRTL, ^†^
*p* < 0.05 AE vs. CTRL-ABZ, and ^‡^
*p* < 0.05 AE vs. AE-ABZ. Values are expressed as mean ± SD (nM).

Compound	CTRL (n = 6) (nM Mean ± SD)	AE (n = 6) (nM Mean ± SD)	AE-ABZ (n = 5) (nM Mean ± SD)	CTRL-ABZ (n = 6) (nM Mean ± SD)
***Unconjugated***
CA	2960 ± 1586	753 ± 761 *^,†^	2218 ± 2861	2760 ± 1521
CDCA	305 ± 148	70 ± 34 *^,†^	237 ± 200	365 ± 346
ωMCA	3074 ± 1654	489 ± 192 *^,†^	1744 ± 1467	2522 ± 741
αMCA	554 ± 385	44 ± 37 *^,†^	287 ± 308	463 ± 399
βMCA	4747 ± 3497	1080 ± 899 *^,†^	4452 ± 4387	4131 ± 2559
UDCA	590 ± 358	122 ±81 *^,†^	784 ± 811	686 ± 549
HDCA	245 ± 95	29 ± 18 *^,†^	174 ± 137	263 ± 91
DCA	917 ± 261	374 ± 146 *^,†^	1217 ± 939	1176 ± 654
12oxoLCA	15 ± 6	5 ± 1 *^,‡^	17 ± 13	14 ± 7
7oxoDCA	686 ± 441	93 ± 74 *^,†^	443 ± 430	609 ± 305
AlloCA	122 ± 45	18 ± 13 *^,†^	63 ± 73	119 ± 52
**Total unconjugated**	**14,215 ± 7740**	**3078 ± 2082 ***	**11,636 ± 11,085**	**13,107 ± 6512**
***Taurine-conjugated***
TCA	345 ± 173	920 ± 620 ^†^	305 ± 282	232 ± 110
TCDCA	27 ± 29	40 ± 23	19 ± 8	22 ± 19
TωMCA	970 ± 228	642 ± 208	607 ±446	822 ± 236
TαMCA	253 ± 104	325 ± 188	329 ± 372	244 ± 158
TβMCA	606 ± 285	1396 ± 916 ^†^	578 ± 596	398 ± 187
TUDCA	110 ± 37	120 ± 33	132 ± 80	125 ± 36
TDCA	81 ± 43	137 ± 69	108 ± 72	80 ± 39
**Total taurine-conjugated**	**2392 ± 755**	**3578 ± 1820**	**2078 ± 1787**	**1924 ± 685**
**Total bile acids**	**16,607 ± 8135**	**6656 ± 3172**	**13,714 ± 12,722**	**15,031 ± 6963**

**Table 2 metabolites-11-00442-t002:** Effects of *E. multilocularis* infection and ABZ treatment on bile acids in liver tissue of mice. The liver samples were collected from non-infected control (CTRL, n = 6), *E. multilocularis*-infected (AE, n = 6), *E. multilocularis*-infected and ABZ-treated (AE-ABZ, n = 5, one outlier with aberrant concentrations was removed), and non-infected and ABZ-treated control mice (CTRL-ABZ, n = 5, one outlier with aberrant concentrations was removed),). * *p* < 0.05 AE vs. CRTL, ^†^
*p* < 0.05 AE vs. CTRL-ABZ, and ^‡^
*p* < 0.05 AE vs. AE-ABZ. Values are expressed as mean ± SD (pg/mg tissue). ND: Not-detected.

Compound	CTRL (n = 6) (pg/mg Tissue Mean ± SD)	AE (n = 6) (pg/mg Tissue Mean ± SD)	AE-ABZ (n = 5) (pg/mg Tissue Mean ± SD)	CTRL-ABZ (n = 5) (pg/mg Tissue Mean ± SD)
***Unconjugated***
CA	8520 ± 9256	151 ± 33 ^†^	2464 ± 3418	1615 ± 631
CDCA	42 ± 31	ND ^‡^	74 ± 63	43 ± 19
ωMCA	1893 ± 1839	182 ± 43 ^†^	964 ± 1010	768 ± 260
αMCA	1627 ± 1627	79 ± 33 ^‡^	1466 ± 2108	542 ± 122
βMCA	4583 ± 3791	2312 ± 1080	6533 ± 5483	3748 ± 1778
UDCA	88± 74	ND ^†,‡^	127 ± 90	107 ± 46
HDCA	50 ± 42	ND ^†,‡^	46 ± 9	43 ± 9
DCA	40 ± 22	14 ± 2 *	26 ± 8	30 ± 11
7oxoDCA	1887 ± 2131	ND *^,†^	519 ± 1041	120 ± 43
**Total unconjugated**	**18,734 ± 18,314**	**2770 ± 1142**	**12,220 ± 13,172**	**7016 ± 1726**
***Conjugated***
TCA	93,465 ± 58,895	53,864 ± 22,985	91,956 ± 95,949	47,978 ± 8176
TCDCA	8584 ± 7344	1980 ± 567	12,736 ± 16,372	5275 ± 2482
TωMCA	43,450 ± 33,129	15,090 ± 5925	23,250 ± 8736	25,676 ± 6107
TαMCA	26,984 ± 21,698	7092 ± 2735 ^‡^	25,708 ± 21,439	14,464 ± 5161
TβMCA	102,187 ± 65,883	84,403 ± 38,111	93,075 ± 51,320	67,780 ± 20,851
TUDCA	12,737 ± 10,579	2449 ± 815 ^‡^	15,399 ± 18621	6981 ± 1806
TDCA	8091 ± 5150	2602 ± 1044	6989 ± 6001	4148 ± 1787
TLCA	319 ± 188	83 ± 35	406 ± 353	275 ± 116
T7oxoLCA	316 ± 377	256 ± 273	462 ± 974	63 ± 61
GCA	264 ± 225	112 ± 27	387 ± 427	157 ± 30
**Total conjugated**	**296,395 ± 198,074**	**167,929 ± 67,355**	**270,370 ± 216,375**	**172,798 ± 39,994**
**Total bile acids**	**314,865 ± 216,097**	**170,587 ± 67,462**	**282,203 ± 229,000**	**179,657 ± 39,812**

**Table 3 metabolites-11-00442-t003:** Increased ratio of taurine-conjugated to unconjugated bile acids in AE. The ratio of total taurine-conjugated to total unconjugated bile acids as well as the ratios of TCA/CA, TαMCA/αMCA, and TβMCA/βMCA were calculated for the four different treatment groups in the serum and liver tissue samples. Non-infected control (CTRL, n = 6), *E. multilocularis*-infected (AE, n = 6), *E. multilocularis*-infected and ABZ-treated (AE-ABZ, n = 5, one outlier with aberrant concentrations was removed), and non-infected and ABZ-treated control mice (CTRL-ABZ, n = 6 in serum and n = 5 in liver tissue due to exclusion of an outlier). Values are expressed as mean ± SD. * *p* < 0.05 AE vs. CTRL, ^†^
*p* < 0.05 AE vs. CTRL-ABZ, and ^‡^
*p* < 0.05 AE vs. AE-ABZ.

Serum	CTRL (n = 6) (mean ± SD)	AE (n = 6) (mean ± SD)	AE-ABZ (n = 5) (mean ± SD)	CTRL-ABZ (n = 6) (mean ± SD)
Total taurine-conjugated/total unconjugated	0.20 ± 0.09	1.38 ± 0.86 *^,†,‡^	0.17 ± 0.07	0.27 ± 0.23
TCA/CA	0.15 ± 0.14	1.63 ± 1.33 *^,†,‡^	0.25 ± 0.26	0.09 ± 0.02
TαMCA/αMCA	0.72 ± 0.63	9.17 ± 5.74 *^,†,‡^	1.50 ± 1.04	0.58 ± 0.17
TβMCA/βMCA	0.17 ± 0.10	1.63 ± 1.18 *^,†,‡^	0.21 ± 0.19	0.12 ± 0.09
**Liver tissue**	**CTRL (n = 6)** (mean ± SD)	**AE (n = 6)** (mean ± SD)	**AE-ABZ (n = 5)** (mean ± SD)	**CTRL-ABZ (n = 5)** (mean ± SD)
Total taurine-conjugated/total unconjugated	46 ± 53	67 ± 33	26 ± 5	26 ± 9
TCA/CA	156 ± 312	359 ± 146 ^†^	50 ± 14	33 ± 13
TαMCA/αMCA	47 ± 53	104 ± 55	27 ± 9	29 ± 15
TβMCA/βMCA	33 ± 23	41 ± 22	16 ± 4	21 ± 11

## Data Availability

All raw data files to the presented experiments will be available at https://zenodo.org/, accessed on 15 June 2021.

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
