# Peer review of "Impact on Bile Acid Concentrations by Alveolar Echinococcosis and Treatment with Albendazole in Mice"

_metabolites, 2021, doi:10.3390/metabo11070442_

Round 1

Reviewer 1 Report

the paper has been corrected by the authors. For me is ok!

Author Response

Response to reviewers

We thank the editor and reviewers for their supportive comments and the suggestions for improvement of the manuscript. The issues raised by the reviewers were addressed as follows:

#Reviewer 1

the paper has been corrected by the authors. For me is ok!

We thank the reviewer for the kind comment.

Reviewer 2 Report

Detailed quantitative analysis of bile acids and salts in serum and liver samples from the mice infected with alveolar echinococcosis (AE) and treated with benzimidazole albendazole (ABZ) was conducted in this study. The expression levels of the enzymes and transporters in bile acid synthesis and turnover were further examined. Altered ratio of taurine-conjugated/total unconjugated bile acids was observed after the AE infection and was further explained by the changes in enzymes and transporters.

---The study has clear merits in the thoroughness of quantitative and biochemical analysis. However, the significance of this study might be limited by the strength and relevance of its experimental animal model, in which the infection occurred in the peritoneal cavity, instead of the liver. As stated in the Results, no clear pathological damage to the liver was observed in the infected mice, except the infiltration of inflammatory cells and the elevation of pro-inflammatory cytokines. Therefore, the observed changes in bile acid profile as well as the inhibition of metabolizing enzymes and transporters were likely the consequences of general inflammation responses triggered by the AE infection. Some sentences in the Discussion have mentioned this mechanism, but they did not address whether the observed changes could be AE specific. Please consider expanding the discussion on the values of bile acid changes as potential biomarkers of AE infection and the pros and cons of this animal model. Also, I would like to suggest a minor revision in the title to reflect the nature of animal model, such as "Impact on Bile Acid Homeostasis by intraperitoneal infection of Alveolar Echinococcosis and Albendazole treatment in Mice". 

---A careful proof on the values in the tables might be needed. The values of omega-MCA as 1893 ± 1839 and alpha-MCA as1627 ± 1627 look odd.

Author Response

Response to reviewers

We thank the editor and reviewers for their supportive comments and the suggestions for improvement of the manuscript. The issues raised by the reviewers were addressed as follows:

 #Reviewer 2

Detailed quantitative analysis of bile acids and salts in serum and liver samples from the mice infected with alveolar echinococcosis (AE) and treated with benzimidazole albendazole (ABZ) was conducted in this study. The expression levels of the enzymes and transporters in bile acid synthesis and turnover were further examined. Altered ratio of taurine-conjugated/total unconjugated bile acids was observed after the AE infection and was further explained by the changes in enzymes and transporters.

---The study has clear merits in the thoroughness of quantitative and biochemical analysis. However, the significance of this study might be limited by the strength and relevance of its experimental animal model, in which the infection occurred in the peritoneal cavity, instead of the liver. As stated in the Results, no clear pathological damage to the liver was observed in the infected mice, except the infiltration of inflammatory cells and the elevation of pro-inflammatory cytokines. Therefore, the observed changes in bile acid profile as well as the inhibition of metabolizing enzymes and transporters were likely the consequences of general inflammation responses triggered by the AE infection. Some sentences in the Discussion have mentioned this mechanism, but they did not address whether the observed changes could be AE specific. Please consider expanding the discussion on the values of bile acid changes as potential biomarkers of AE infection and the pros and cons of this animal model. Also, I would like to suggest a minor revision in the title to reflect the nature of animal model, such as "Impact on Bile Acid Homeostasis by intraperitoneal infection of Alveolar Echinococcosis and Albendazole treatment in Mice". 

We appreciate the reviewer’s comment. As suggested, we have modified the discussion to address the suggested points. We are aware this animal model does not reproduce completely the human pathology of AE infection, but our used model clearly reproduces the inflammatory component of this disease. The downsides of an oral model were already discussed in section 2.1, but we emphasized the pros and cons in the discussion. By using the intraperitoneal infection model, we were able to mimic a chronic infection and the treatment as for the chronic human AE. For future studies, already under investigation, an oral infection would reproduce the human natural infection and probably the lesions in the liver would be observed. The oral infection model would require a higher level of biosafety and have other drawbacks such as the viability of the E. multilocularis eggs, or the difficulties on observance of the treatment effect.

---A careful proof on the values in the tables might be needed. The values of omega-MCA as 1893 ± 1839 and alpha-MCA as1627 ± 1627 look odd.

We thank the reviewer for the kind comment. We have double checked the data and all the numbers in the Tables are correct.

This manuscript is a resubmission of an earlier submission. The following is a list of the peer review reports and author responses from that submission.

Round 1

Reviewer 1 Report

Gomez et al. reported the variation of bile acids and of some of the enzymes involved in bile acid synthesis and homeostasis in relation to the development of alveolar echinococcosis (EC) and treatment with Albendazole. In general, the paper is well-written, and results are shown clearly.

Regarding LC-MS, it is important to give some other details, for example the transition used for quantification and qualification, in order to confirm what reported in [28]. In supplementary materials is also important to add some information regarding LOD, LOQ and calibration curves.

I recommend the present article for publication.

Author Response

Response to reviewers

We thank the editor and reviewers for their supportive comments and the suggestions for improvement of the manuscript. The revision experiments were performed by C. Gomez; therefore, and in agreement with all authors, she should be listed as sole first-author. We increased the resolution of the figures and uploaded them also separately. The issues raised by the reviewers were addressed as follows:

#Reviewer 1

Gomez et al. reported the variation of bile acids and of some of the enzymes involved in bile acid synthesis and homeostasis in relation to the development of alveolar echinococcosis (EC) and treatment with Albendazole. In general, the paper is well-written, and results are shown clearly.

Regarding LC-MS, it is important to give some other details, for example the transition used for quantification and qualification, in order to confirm what reported in [28]. In supplementary materials is also important to add some information regarding LOD, LOQ and calibration curves.

We thank the reviewer for the suggestion and added a new Supplemental Table S1 with some details regarding the LC-MS method, including quantifier and qualifier MRM transitions, collision energies used, electrospray ionization mode, and retention times of the compounds. The information regarding the calibration curves for the 36 compounds screened, limits of detection, and limits of quantification were described in detail in the referenced manuscript (Gómez et al., Development and Validation of a Highly Sensitive LC-MS/MS Method for the Analysis of Bile Acids in Serum, Plasma, and Liver Tissue Samples. Metabolites 2020, 10, 282, doi:10.3390/metabo10070282).

Reviewer 2 Report

I have now reviewed the research entited:  Impact on serum bile acid concentrations by alveolar echino-coccosis and treatment with albendazole in mice. I would like to congratulate the authors for their sound research work.

  • What is the preference of multilocularis infections in humans and how much tis is attributed to domestic dogs and cats?
  • What are the effects of stress hormones (Cortisol) and their status during infestation?
  • Does the increased synthesis of bile acids in these mice disturb the steroids biosynthetic pathways? And if yes, what are the consequences?
  • The SDs for individual bile acids as well as the sums of taurine-conjugated, unconjugated and total bile acids analyzed by LC/MS/MS are too high for some bile acids. Has this method been validated fully? Have the authors compared the total and conjugated levels with the routine biochemical methods?
  • Have the authors measured the tissue bile acids to compare to the levels of circulating plasma levels? It is very important to see the intra cellular bile acids levels.
  • In this mouse model of AE, what are the clinical side effects of ABZ treatment and also in the control group?
  • Increased ratio of taurine-conjugated to unconjugated bile acids in AE seems to be a good indicater. Have the authors calculated this ratio using the conventional method and found an advantage for the LC/MS method over the routine analysis?
  • I agree with the authors that the ratio seems to be a good evidence for screening as diagnostic markers to distinguish different liver diseases and monitor disease progression and therapeutic efficacy.

Author Response

Response to reviewers

We thank the editor and reviewers for their supportive comments and the suggestions for improvement of the manuscript. The revision experiments were performed by C. Gomez; therefore, and in agreement with all authors, she should be listed as sole first-author. We increased the resolution of the figures and uploaded them also separately. The issues raised by the reviewers were addressed as follows:

#Reviewer 2

I have now reviewed the research entited:  Impact on serum bile acid concentrations by alveolar echino-coccosis and treatment with albendazole in mice. I would like to congratulate the authors for their sound research work.

What is the preference of multilocularis infections in humans and how much tis is attributed to domestic dogs and cats?

In Western and Central Europe, 0.3 to 3 per 1,000,000 inhabitants get infected with Emultilocularis annually and case numbers are on the rise. E. multilocularis typically undergoes a predator-prey life cycle, involving a carnivorous definitive host and mostly small rodents as intermediate hosts. Humans, as well as dogs, beavers and captive monkeys are considered accidental intermediate hosts. Adult stages of the tapeworm are present in the intestine of red foxes, arctic foxes, coyotes, raccoon dogs, wolves, or domestic dogs (B. Lundström-Stadelmann et al., Food and Waterborne Parasitology 12 (2019) e00040 https://doi.org/10.1016/j.fawpar.2019.e00040).

Although not entailing large numbers of domestic dogs, cases of canine AE have been reported on a regular base from Central Europe and from Canada since the past three decades. Domestic dogs with infections have been found in several areas and may play an important role for parasite transmission to humans; however, they are probably not significant for maintaining the cycle in Central Europe. Cats, with low worm burdens and correspondingly low egg excretion, are probably of negligible zoonotic significance in the maintenance of the E. multilocularis life cycle (Deplazes, et al., Advances in Parasitology, 2017, 95;315-493, https://doi.org/10.1016/bs.apar.2016.11.001).

Based on recently improved diagnostic strategies, E. multilocularis infections have been diagnosed in dogs and cats in Switzerland, Czech Republic, Germany and France. Prevalence of <1.5% was recorded in privately owned rural and urban pet dogs, but higher prevalences of 3–8% were found in dogs with free access to rodent habitats such as farm dogs and hunting dogs.

The prevalence of E. multilocularis in domestic or wild cats in Europe (determined by necropsy) ranged between 0% and 5.5%.

We modified the introduction by adding the following information in the first paragraph: “In the Northern Hemisphere, 0.3 to 3 per 1,000,000 inhabitants get infected annually with E. multilocularis, with increasing numbers. Adult stages of the tapeworm occur mainly in the intestine of red and arctic foxes, although domestic dogs and cats can also act as definitive hosts. A prevalence <1.5% was found in privately owned rural and urban pet dogs, wheras it was 3–8% in dogs with free access to rodent habitats such as farm dogs and hunting dogs.”

What are the effects of stress hormones (Cortisol) and their status during infestation?

This is a good question. We did not quantify corticosterone or other steroids in this study due to limited sample availability. Also, the design of this study was not intended to address this. However, we agree with the reviewer that this is an interesting issue to be investigated in a follow-on study. Elevated corticosterone in mice, or elevated cortisol in human as well as pharmacologic glucocorticoid treatment are expected to result in immune suppression, which likely leads to more severe infection and disease progression. A possible reason for the increased number of patients with AE in the last 3 decades may be due to a higher number of immune suppressed individuals. Whether elevated glucocorticoid levels due to stress contribute to higher infection rate remains unclear. A possible link between glucocorticoids and AE severity and progression will need to be studied in appropriate mouse models, for example by pharmacological administration, and in suitable human cohorts, testing for correlation of glucocorticoid concentrations and disease parameters.

Does the increased synthesis of bile acids in these mice disturb the steroids biosynthetic pathways? And if yes, what are the consequences?

Bile acid synthesis was not increased in the AE infected mice but unconjugated bile acids were lower and taurine-conjugated bile salts slightly higher in serum and unchanged or tending lower in liver tissue, probably due to altered transport. Whether this affects steroid biosynthesis in adrenals or gonads is unknown, also whether hepatic steroid degradation might be altered. We observed decreased Akr1d1 in AE infected livers. Besides its role in bile acid synthesis, AKR1D1 is involved in the degradation of androgens and glucocorticoids. However, whether this affects steroid homeostasis will require quantification of plasma steroid levels. Such a study should then also include expression of 5alpha-reductase and CYP3A for glucocorticoid catabolism.

The SDs for individual bile acids as well as the sums of taurine-conjugated, unconjugated and total bile acids analyzed by LC/MS/MS are too high for some bile acids. Has this method been validated fully? Have the authors compared the total and conjugated levels with the routine biochemical methods?

We appreciate the reviewer’s comment. The method used in the present study to quantify bile acids was developed and validated for the analysis of 36 compounds in serum, plasma and liver tissue. The details of the validation were described in detail earlier (Gómez et al., Development and Validation of a Highly Sensitive LC-MS/MS Method for the Analysis of Bile Acids in Serum, Plasma, and Liver Tissue Samples. Metabolites 2020, 10, 282, doi:10.3390/metabo10070282). We now also added additional information regarding the LC-MS method, including quantifier and qualifier MRM transitions, collision energies, electrospray ionization mode, and retention times of the compounds in a new Supplemental Table S1. The information regarding calibration curves, recoveries for the 36 compounds screened, limits of detection and limits of quantification can be found in the referenced method paper.

The SDs for individual bile acids typically are high. This is not an issue of the method but of the physiology. This can be attributed to inter-individual variability but also variability within an animal due to feeding, day time, and physical activity. In the present study, we did not starve the mice for a certain period of time and mice had access to food ad libitum, mimicking a normal situation (with the disadvantage of higher variability). Food consumption is known to affect bile acid release and serum concentrations. Secondary bile acids are typically generated by the gut microbiome. The microbiome displays a considerable inter-animal variability. Also, the number of mice was rather low in this study. However, the observed decrease of all unconjugated bile acids in AE infected mice is striking, and that pattern unlikely would change by including additional animals or having lower variability.

Have the authors measured the tissue bile acids to compare to the levels of circulating plasma levels? It is very important to see the intra cellular bile acids levels.

We agree with the reviewer that this is an important point. Due to limited sample availability we had not done this in a first step. We have now used the last remaining tissue piece to quantify bile acids in liver tissue. We mention effects of AE on liver tissue bile acid profile in the abstract and results section. Figures and Tables have been updated in the manuscript. Table 2 contains the concentrations of bile acids in liver tissue, and the individual data points of bile acids quantified in liver tissue of mice from the four treatment groups are depicted in Supplemental Figure S2.  Bile acid profiles and ratios described previously for serum have also been included and depicted for liver tissue samples (modified Figure 1 and Table 3).

In this mouse model of AE, what are the clinical side effects of ABZ treatment and also in the control group?

The ABZ dose used in this study, 200 mg/kg/d five times per week over 8 weeks, was used in several previous studies and did not show adverse events. Importantly, the ABZ treated animals did not show altered serum bile acid profiles, whereas bile acids in liver tissue tended to be lower which might be a sign of slight disturbance of hepatic function. In a long-term treatment study with ABZ at a dosage of 136.3 mg/kg/d significantly elevated serum levels of ALT, AST and ALP were observed, along with mild pathological changes in the liver of treated mice (Zheng, Q. et al., Chinese Journal of Parasitology and Parasitic Disease 2013;31(3):193-7). We did not measure liver toxicity biomarkers but histopathological analyses did not show any signs of hepatic damage in the ABZ control group. We mention this in the text.

The mode-of-action of ABZ is explained by strong binding to the nematode’s tubulin, preventing polymerization. Intestinal cells of the nematode are particularly affected, resulting in a loss of absorptive function, causing death. Oral adminstration in mice, rats, hamsters, guinea pigs and rabbits was generally found to have low toxicity (https://www.ema.europa.eu/en/documents/mrl-report/albendazole-summary-report-3-committee-veterinary-medicinal-products_en.pdf).

Increased ratio of taurine-conjugated to unconjugated bile acids in AE seems to be a good indicater. Have the authors calculated this ratio using the conventional method and found an advantage for the LC/MS method over the routine analysis?

There are different enzymatic assays on the market for the detection of total bile acids (TBA) in different matrices such as plasma, serum, tissue homogenates or cell lysates. These assays are based on absorbance measurements at different wavelength, depending on the kit. They usually require 10 to 50 µL of sample volume and the concentration range measured is above 1 to 10 µM, depending on the test brand. Importantly, none of these kits can differentiate between conjugated and unconjugated bile acids and therefore no ratio can be analyzed. The conventional kits are suitable to detect robust changes in total bile acids such as present in cholestasis. Subtler differences require mass-spectrometry based methods.

I agree with the authors that the ratio seems to be a good evidence for screening as diagnostic markers to distinguish different liver diseases and monitor disease progression and therapeutic efficacy.

We thank the reviewer for the kind comment.

Reviewer 3 Report

The authors conducted rather detailed bile acid analysis on serum samples from the mice infected with Alveolar echinococcosis as well as the ones with drug treatment. After observing the changes in both quantity and composition of serum bile acids, the expression levels of related genes and proteins in bile acid biosynthesis and transport were examined. Reasonable explanation on the observed changes has been proposed to link the infection-induced inflammation and the changes in bile acids. Overall, this is a manuscript based on a sound experimental animal model and detailed sample analysis. Nevertheless, I have a few suggestions for the authors to consider:

  1. Considering hepatic bile acids are the source of serum bile acids, hepatic bile acid data is a missing piece in this study. Since the liver samples have been collected from this study, it is desirable to have this data to connect the serum data with the bile acid-related gene and protein expression in the liver.
  2. Figure 1 should be placed as a summary figure in the Discussion, instead of in the Introduction, because it contains all the targets in this study as well as the potential mechanism.  Once it is placed into the Discussion, the results of this study, including the changes in bile acids, genes, and proteins, should all be included in the figure. The major content from line 62 to line 93 could also be transferred to the Discussion.
  3.  If available, the authors may consider to present the results of liver function assays, such as transaminase activity, alkaline phosphatase, and albumin.

Author Response

Response to reviewers

We thank the editor and reviewers for their supportive comments and the suggestions for improvement of the manuscript. The revision experiments were performed by C. Gomez; therefore, and in agreement with all authors, she should be listed as sole first-author. We increased the resolution of the figures and uploaded them also separately. The issues raised by the reviewers were addressed as follows:

#Reviewer 3

The authors conducted rather detailed bile acid analysis on serum samples from the mice infected with Alveolar echinococcosis as well as the ones with drug treatment. After observing the changes in both quantity and composition of serum bile acids, the expression levels of related genes and proteins in bile acid biosynthesis and transport were examined. Reasonable explanation on the observed changes has been proposed to link the infection-induced inflammation and the changes in bile acids. Overall, this is a manuscript based on a sound experimental animal model and detailed sample analysis. Nevertheless, I have a few suggestions for the authors to consider:

  1. Considering hepatic bile acids are the source of serum bile acids, hepatic bile acid data is a missing piece in this study. Since the liver samples have been collected from this study, it is desirable to have this data to connect the serum data with the bile acid-related gene and protein expression in the liver.

We agree with the reviewer. Due to limited sample availability we had not done this in a first step. As suggested we have now quantified bile acids in liver tissue. We mention effects of AE on liver tissue bile acid profile in the abstract and results section. Figures and Tables have been updated in the manuscript. Table 2 contains the concentrations of bile acids in liver tissue, and the individual data points of bile acids quantified in liver tissue of mice from the four treatment groups are depicted in Supplemental Figure S2. Bile acid profiles and ratios described previously for serum have also been included and depicted for liver tissue samples (modified Figure 1 and Table 3).

  1. Figure 1 should be placed as a summary figure in the Discussion, instead of in the Introduction, because it contains all the targets in this study as well as the potential mechanism.  Once it is placed into the Discussion, the results of this study, including the changes in bile acids, genes, and proteins, should all be included in the figure. The major content from line 62 to line 93 could also be transferred to the Discussion.

As suggested we now placed the original Figure 1 in the Discussion section (new Figure 5) and modified it to include the main findings (now also AKR1D1 and CYP7A1 as well as effect on bile acids). We assume the observed changes are mainly caused by the hepatic inflammation. As the role of FXR was not investigated, we removed it from the Figure.

We prefer to leave the information of lines 62-93 in the introduction to allow readers not familiar with bile acid homeostasis to better follow the selection of enzymes and transporters analysed and the role they have in regulating bile acids.

  1.  If available, the authors may consider to present the results of liver function assays, such as transaminase activity, alkaline phosphatase, and albumin.

Unfortunately, we now used up all samples (serum and liver tissue) and AST, ALT or albumin have not been measured. The histological analysis of the liver tissue, however, did not show any lesions or overt hepatic damage in all groups, indicating that neither AE nor ABZ or the combination led to gross hepatic damage.